# Assessing Autistic Traits in Toddlers Using a Data-Driven Approach with DSM-5 Mapping

**DOI:** 10.3390/bioengineering10101131

**Published:** 2023-09-27

**Authors:** Neda Abdelhamid, Rajdeep Thind, Heba Mohammad, Fadi Thabtah

**Affiliations:** 1Abu Dhabi School of Management, Abu Dhabi P.O. Box 6844, United Arab Emirates; 2Manukau Institute of Technology, Auckland 2023, New Zealand; thin34@manukaumail.com; 3Higher Colleges of Technology, Abu Dhabi P.O. Box 25026, United Arab Emirates; hmohammad@hct.ac.ae; 4ASDTests, Auckland 0610, New Zealand; fadi@asdtests.com

**Keywords:** autism, autistic traits, classification, data analysis, feature selection, machine learning

## Abstract

Autistic spectrum disorder (ASD) is a neurodevelopmental condition that characterises a range of people, from individuals who are not able to speak to others who have good verbal communications. The disorder affects the way people see, think, and behave, including their communications and social interactions. Identifying autistic traits, preferably in the early stages, is fundamental for clinicians in expediting referrals, and hence enabling patients to access to required healthcare services. This article investigates various ASD behavioral features in toddlers and proposes a data process using machine-learning techniques. The aims of this study were to identify early behavioral features that can help detect ASD in toddlers and to map these features to the neurodevelopment behavioral areas of the Diagnostic and Statistical Manual of Mental Disorders (DSM-5). To achieve these aims, the proposed data process assesses several behavioral features using feature selection techniques, then constructs a classification model based on the chosen features. The empirical results show that during the screening process of toddlers, cognitive features related to communications, social interactions, and repetitive behaviors were most relevant to ASD. For the machine-learning algorithms, the predictive accuracy of Bayesian network (Bayes Net) and logistic regression (LR) models derived from ASD behavioral data subsets were consistent pinpointing to the suitability of ML techniques in predicting ASD.

## 1. Introduction

ASD is a neurodevelopmental condition that affects people’s perception of the world [1]. According to the [2], one in every 100 children is estimated to have ASD. Zeidan et al., (2022) [3] stated that there are more male cases of ASD than female cases. In the USA, the average age for detecting autism is four years, and families sometimes need to wait for 13 months after the first screening for their children for further assessments. According to Kosmicki et al., (2015) [4], this period can be longer for minority communities or for people of lower socioeconomic status. A study by [5] found that the average age of patients diagnosed with ASD is approximately 60 months.

In spite of the progress made in the early detection of autism, a clear decline was noticed during the COVID-19 pandemic, according to a report of the Centers for Disease Control and Prevention Report (CDC, 2023) [6]. Each person with autism is affected in a different way. People with autism can also exhibit other neurodevelopmental disorders, such as attention deficit hyperactivity disorder (ADHD) [1,7]. 

Currently, with the rapid developments in computer applications, and data analytics, which have enhanced the collection and availability of data [8], the healthcare sector is heavily reliant on data-based systems in guiding decisions. Medical-related datasets that are processed using artificial intelligence (AI) and data analytics techniques can be vital in the discovery of useful insights that can be utilized in diagnosing patients and improving their treatments [9]. The applications of data-driven approaches to assist in improving the ASD detection rate and to determine influential ASD traits have gained attention in the medical community [10,11,12,13,14,15,16]. However, little work has been focused on detecting ASD traits in toddlers [10,11,17,18] to investigate early indicators of ASD or the relationship of such indicators to the neurodevelopment behavioral areas described in the Diagnostic and Statistical Manual of Mental Disorders (DSM-5) [19]. 

This research proposes a data-driven process to improve the screening of ASD in toddlers, applying machine-learning algorithms and using data collected by AutismAI system [20]. AutismAI embeds medical ASD-screening questionnaires, such as the short version of the Quantitative Checklist for Autism in Toddlers (Q-CHAT-10) [21], and others. The proposed data process can be used to determine a minimal set of dissimilar features to build classification models for assessing ASD. By identifying these features from ASD–toddler data, medical professionals can accurately carry out ASD screening in a cost-effective manner, using models developed by classification algorithms, and thereby reach clinical screening decisions as soon as these behavioral traits are identified during a clinical session. 

Unlike most current data-driven methods that have been used in studying the behavioral features of ASD in toddlers such as the methods used by [17,22], and others, the current study’s main contribution is not only to consider the relationships between behavioral features of the ASD-screening methods used for assessment, but also to consider the correlations between the features themselves. Therefore, we can identify possible redundant features in ASD-screening methods. More essentially, this study maps the relationship between the ASD features with their neurodevelopment behavioral areas according to the DSM-5, opening discussions about which criteria are most vital in early diagnosis and which criteria pertain to the most behavioral features within the ASD diagnostic tools. 

This paper is organized as follows: Section 2 provides a literature review and sets out works related to ASD detection in toddlers. Section 3 outlines the methodology followed. In Section 4, an analysis of the findings of the experiment are described and discussed. Finally, in the conclusion section, a summary of the results and the study’s experiment limitations are outlined.

## 2. Literature Review 

### Related Works

Zhu et al., (2023) [10] proposed a machine-learning approach to detect ASD in toddlers, using audio and a face-based detection system to assess response-to-name (RTN) as an ASD indicator. They compared human-based ratings and computer-based ratings for RTN. In that study, approximately 125 toddlers aged between 16 months and 30 months participated in the experiment, 61 of whom had been diagnosed with ASD, 31 of whom were diagnosed with developmental delays, and 33 of whom were diagnosed with typical delays. Caregivers and clinical evaluators were involved in the study. In the proposed system, the main features used to detect RTN were response latency, response duration, and head pose. For facial feature detection, a DLiB library was used with deep-learning method [23]. The human-rated approach and the computer-based approach were compared using area-under-curve (AUC) and accuracy measurements. According to the authors, both approaches reported consistent results. The AUC measurements reported a significant difference between human-based ratings and computer-based ratings (0.91 and 0.81, respectively). The other performance measurements reported no significant difference. The computer-based detection of RTN reported sensitivity, specificity, and accuracy of 80.0%, 69.8%, and 74.8%, respectively. For the human-based rating approach, the specificity was 82.5%, the sensitivity was 83.3%, and the accuracy was 82.9%. The authors emphasized that the use of machine-learning approaches for the early detection of ASD was promising; however, relying on a single factor, such as RTN, was not sufficient, and the dynamics and the evolution of ASD behavior required further consideration in future studies.

In 2023, [10] relied on a single predictor (response-to-name) to assess ASD in toddlers, using a machine-learning approach. In 2022, [11] chose to detect ASD in children by collecting their dataset via a semi-structured experiment using audio and video recordings. They considered different variables, including demographic information, medical diagnoses, and healthcare service procedures that were collected and extracted from individual cases. The dataset used was the IBM MarketScan Health Claims Database, which covered claims from 2005 to 2016. Their study included 38,576 observations (12,743 of which were ASD-diagnosed and 25,833 of which were non-ASD-diagnosed). The chronological age of the ASD cohort ranged from 18 months to 30 months, while the ages of the non-ASD cohort extended to 60 months. The researchers applied LR and random forest (RF) algorithms to discriminate between ASD and non-ASD cases. The RF algorithm reported better performance than the LR algorithm (AUCROC: 0.78 and 0.76, respectively), with specificity of 93% at the age of 24 months. In addition, the authors determined that better detection was found as the age of the toddlers increased. This was expected, as ASD symptoms, in particular the behavioral and social symptoms, are more obvious and pronounced as the age of a toddler increases [24]. The authors stated that reliance on on medical claims and insurance claims was not sufficient. They recommended that in future studies, the integration of data related to ASD developmental screening results and behavior data would be beneficial. 

In 2022, [12] developed a system to pre-diagnose autism that can be used by caregivers, parents, and autistic people (children and adults). Their system uses questions adopted from medical questionnaires, such as Q-CHAT-10 and the various versions of AQ-10. Their system consists of a two-layer convolutional neural network (CNN) with 32 and 64 filters, respectively, and a dataset covering different age categories (toddlers, adolescents, and adults) that was collected by [25,26] with 6075 data observations. The reported performance measurements for accuracy, sensitivity, and specificity were 95.53%, 97.63%, and 98.63%, respectively. The developed CNN outperformed the other machine-learning algorithms, including the decision tree algorithm, the rule induction algorithm, and the Bayes Net algorithm. The authors recommended investigating other machine-learning approaches, such as deep learning, using more complex features such as videos and images to detect ASD. They also recommended using cluster analysis to recognize and tune the treatment strategies for different ASD cases.

In 2022, [13] suggested using a machine-learning approach to evaluate the predictive performance of the ASD-screening process. They suggested a system with two phases. The first phase was a pre-diagnostic phase, where the input dataset was clustered based on independent features related to communication, repetitive traits, and social traits, using a self-organizing-map (SOM). A new class label was created and compared with the existing class label, to refine the dataset and reduce the bias of the screening system’s assigned class. In this phase, 85% accuracy was achieved. In the second phase (the classification phase), a refined dataset was used and the prediction of ASD was remeasured. RF and naïve Bayes (NB) algorithms were used for classification. Approximately 2000 data instances are used to assess the derived models. This experiment provided good performance results in terms of accuracy, precision and recall by both classifiers (NB: 93%, 93%, and 94%; RF: 96%, 96%, and 97%). The RF classifier reported better performance results than the NB classifier. The evaluation of predictive models of ASD-detection systems using unsupervised learning is a creative approach to improving the quality of the machine-learning screening systems of ASD and reducing any bias related to medical screening. 

In 2020, [22] discussed the issue of imbalanced class labels and how prediction performance can vary. They used the NB algorithm on nine ASD datasets, with several resampling techniques and different class label ratios, with 100 runs for each. The original dataset used was generated by the ASDTest system [25]. The authors used 1000 observations (975 with no ASD traits and 25 with ASD traits). The resampling techniques used with the NB classifier were the synthetic minority oversampling technique (SMOTE), the random oversampling (ROS) technique, and the random undersampling (RUS) technique. The SMOTE with the NB classifier reported the best ROC, while the NB with RUS reported the lowest performance results. Using the resampling techniques with NB helped to improve the predictive performance in the imbalanced ASD datasets. That research was one of the few studies to investigate the imbalance class label in ASD screening, together with those of, [13,15], and [16].

Washington et al., (2019) [27] explored feature redundancy by performing filter, wrapper, and embedded-feature selection analyses. The data used were aggregated from seven different sources consisting of 16,527 children/adolescents and the completion of with social responsiveness scale (SRS) [28] child/adolescent questionnaires. The SRS is a 65-item questionnaire that is completed by a caregiver about a child. Univariate filter feature selection was applied by the authors to measure the correlation between each feature and the outcome (class). They incorporated the recursive-feature-elimination (RFE) wrapper method, repeatedly removed the weakest feature, and created predictive models until the number of features achieved the desired performance. A support vector machine (SVM) algorithm was applied at each step of the RFE procedure to remove a single feature. During the classification step, principal component analysis (PCA), t-distributed stochastic neighbor embedding (t-SNE), and denoising autoencoder were used with a multi-layer perceptron (MLP) classifier to process the top-ranking items and to derive a classification model with an AUC of 92%.

In 2020, [29] determined that early detection of the symptoms of ASD can improve the quality of life of the diagnosed individuals. The authors used integrated data from the University of California, Irvine (UCI) machine learning repository, which consisted of three datasets with 20 common attributes. Data were cleaned for missing values and outliers. During the experiments, training and testing data settings in the ratio of 80:20 were applied, with cross-validation. SVM, and NB algorithms were used for classification, along with CNN, LR, and K-nearest neighbor. The derived predictive accuracy in detecting ASD using the CNN algorithm reached 98.30%. 

In 2020, [30] suggested using feature selection with classification techniques to decrease data dimensionality and to choose only relevant features to enhance classification accuracy. They explained that various feature-selection methods can be used in ASD research and can help in improving the efficiency and the performance of machine-learning classification algorithms such as the flat, the streaming, and the structured feature engineering approaches. According to their recommendation, feature selection in ASD research requires additional investigation, due to its essential role in the pre-processing phase. 

Kosmicki et al., (2015) [4] attempted the development of a more accurate method of fast detection of ASD than the current standard methods. They used machine-learning techniques to evaluate the clinical assessment of ASD, using the Autism Diagnostic Observation Schedule (ADOS) [31] to test whether a smaller subset of behaviors could differentiate between children who exhibited ASD traits and those who did not. The ADOS presents behavioral observations in a clinical setting and comprises four modules with different levels of cognitive functioning. The authors estimated that 27% of individuals are undiagnosed at the age of 8 years. Eight machine-learning algorithms were used, with feature selection, to process ADOS-related data. The results showed that the number of behavior traits was reduced from 28 to 9, at least for ADOS Module 2, and from 28 to 12 for ADOS Module 3. The LG algorithm showed good performance results when processing 9-trait data of Module 2, with 98.81% and 89.39% specificity. The SVM algorithm that was used when processing on Module 3 exhibited 97.71% and 97.20% specificity (a true-negative rate). In 2015, [4] claimed that these results encouraged the development of screening-based instruments for ASD detection and mobile-health approaches that enable individuals to receive necessary care. 

In 2019, [32] considered that the revised Modified Checklist for Autism in Toddlers (M-CHAT-R) [33] was good for the initial screening of children for autism, but it required follow-up questions, as the interpretation of its scores by humans can be biased. The authors believed that AI methods could overcome the barriers to ASD screening, such as the use of a feed-forward artificial neural network [fANN] [34]. Hence, processing real data related to M-CHAT could not only be accessible, with low-cost screening, but it could be reliable for rural, minority, and low socioeconomic populations with low education levels. In experimentations, [32] included a total of 16,168 toddlers. The machine-learning technique was applied to the complete dataset and filtered by race, gender, and the parents’ education levels, and then the results were compared. The results produced high rates of correct classification using 18, and 14 attributes for toddlers respectively. The researchers claimed that the machine-learning method was comparable to the M-CHAT-R in accuracy of ASD diagnosis, while using fewer items. 

Rahman et al., (2020) [30] determined that early identification approaches for ASD were limited and that most toddlers were not identified until after the age of four years, even though evidence suggested that early intervention and diagnosis could lead to major developmental improvements for the children. They applied machine-learning methods to electronic medical records (EMRs) to predict ASD early in a child’s life. The data included 1397 ASD children and 94,741 non-ASD children, born between January 1997 and December 2008. Parental sociodemographic information, parental medical history, and prescribed medication data were used to create 89 features for training and testing with various machine-learning algorithms, including multivariate LR, artificial neural networks, and RF. Additionally, 10-fold cross-validation was used to evaluate prediction performance by computing the area under the operating characteristic curve (AUC, C-statistic), sensitivity, specificity, accuracy, false positive rate, and precision. All machine-learning models produced similar performances, with a C-statistic of 0.709, sensitivity of 29.93%, specificity of 98.18%, accuracy of 95.62%, a false positive rate of 1.8%, and precision of 43.35% for the prediction of ASD in the dataset. Table 1 provides a summary of the literature review.

## 3. Methodology 

This research followed the methodology depicted in Figure 1. The dataset used was collected from a mobile application called AutismAI. This dataset consisted of several characterized behavioral features related to Q-CHAT-10, together with three AQ screening methods plus other characteristics that are useful in detecting ASD. The dataset included 2048 data observations and included four age groups: toddlers (aged 12–36 months), children (aged 3–11 years), adolescents (aged 12–16 years), and adults (aged 17 years and over). Our focus was on toddlers only, a subset of 401. The data collection BY AutismAI was conducted after obtaining ethical approval from the host educational institutions. According to the authors, during data collection, there was no direct human-to-human contact; instead, participants such as parents used the system, after agreeing on an electronic consent form that data were to be used for research purposes only and were stored on a secured cloud database [20].

The medical questionnaire used for feature assessment was the short version of the Q-CHAT-10. Baron-Cohen et al., [35] created a checklist for autism in toddlers (CHAT), which is a screening instrument that identifies children who are 18 months of age and who are at risk for autism. Early detection of autism has been rare before the age of three years, as it is an uncommon condition and no specialized screening tools exist. By the age of 18 months, children on the spectrum exhibit an absence of behaviors such as joint attention and pretend play. Joint attention is the ability to establish a shared focus of attention with another person, while pretend play is an imaginary feature attribute involving pretending (by themselves or with another person) and imagining. 

CHAT comprises nine questions that a health professional asks a parent, followed by direct observation of five aspects. The key items are related to pretend play, protodeclarative pointing, following a point, pretending, and producing a point. If the child fails all of these items, the child is predicted to be at the greatest risk of autism. CHAT was tested on a population of 16,000, and the high-risk criteria had a sensitivity of 18%, a specificity of 100%, a positive predictive value of 75%, and a negative predictive value of 99.7% [35]. Q-CHAT originally consisted of 25 questions; it was later reduced by [21] to 10 questions in the Q-CHAT-10 version, to speed up the screening process. Table 2 shows the items/questions of the screening methods considered. For the 10 questions, a score of 1 or 0 is recorded, based on a parent’s answer, If the total score is more than 3, the child is referred for further assessment by health professionals [21].

Figure 2 indicates that male toddlers score higher on Q-CHAT-10 than female toddlers. This indicates that boys have a higher risk of ASD than girls, but this result could also be because there were more male toddlers than girl toddlers who were screened the AutismAI screening app. 

Attributes for id, date, and AutismAgeCategory were removed, as these were irrelevant for the experiment. To conduct further feature selection, as there are potentially four class attributes, the Score, DNNPrediction, and IsASDDignosed attributes were also removed, as they basically represented the same class and could have added extra weight to the model. Finally, the values of the Q1–Q10 attributes were transformed into nominal notations, as the responses to these questions from the Q-CHAT-10 could be either 0 or 1.

We used different feature-selection methods, including Pearson correlation, Relief-f [36] and gain ratio (GR) [37], as these methods are commonly used in medical research and provide dissimilar mathematical approaches to define the relevancy of each feature. GR can be calculated as follows:(1)$$GR\left(att\right)=\frac{IG\left(att\right)}{H\left(att\right)}$$
where *IG*(att) is the information gain (IG) of each attribute and *H*(att) is the entropy of the attribute *att* by contributing to the class. GR is used to reduce bias towards multi-valued attributes, which is one of the main limitations of the IG metric [38].

The Pearson correlation can be calculated as follows:(2)$$PCor=\frac{\sum XY-\frac{\left(\sum X\right)\left(\sum Y\right)}{n}}{\sqrt{\left({\sum X}^{2}-\frac{{\left(\sum X\right)}^{2}}{n}\right)\left({\sum Y}^{2}-\frac{{\left(\sum Y\right)}^{2}}{n}\right)}}$$

The Pearson correlation coefficient defines a linear correlation between two sets of data. It is essentially the covariance of the two variables divided by the product of their standard deviations. The Relief-f method is inspired by instance-based learning and detects features that are statistically relevant to the target variable. It calculates a score for each feature which is applied to a rank and, then, selects top-scoring features for feature selection. Relief-f evaluates the importance of an attribute by repeatedly sampling an instance and considering the value of the given attribute for the nearest instance of the same class and a different class [36]. Relief-f calculates the weight of a feature (w_j_) using Equation (3) [39]:(3)$$w_{j}=\frac{\sum [\left(x_{ij},x_{ih}\right)-\left(x_{ij},x_{im}\right)]}{N}$$
where

*N* is the number of instances in a dataset with M features.

For an instance (x_i_), where x_i_ is a vector x_i_ = (x_i_₁, x_i_₂, .., x_i__m_), where i = 1, 2, .., N:*x*_i__h_ = the instance of the same class as x_i_ (nearest hit neighbor);*x*_i__m_ = the instance of a different class (nearest miss neighbor);δ(x_i__j_, *x*_i__h_) = the difference between the feature j values of x_i_ and its nearest hit neighbor x_i__h_; andδ(*x*_i__j_, *x*_i__m_) measures the difference between the feature j values of x_i_ and its nearest miss neighbor x_i__m_.

For the classification step of the data process, we used two different algorithms—the LR and the Bayes Net. These algorithms were used previously in medical [40,41,42] and they have different learning methods. LR is a classification algorithm that uses a logistic function to describe probabilities of a possible outcome of a single trial. Logistic function is designed for classification and is useful for understanding the effects that various independent variables have on a single outcome variable. LR only works on a binary variable under the assumption that all independent variables are independent of each other.

LR can be calculated as follows:(4)$$Log\left(\frac{p}{1-p}\right)=\beta 0+\beta 1x1+\beta 2x2$$

The Bayes Net algorithm, derived from the Bayes theorem, is a classification technique that uses a probabilistic graphical model to represent features and that can be used on a variety of tasks, including prediction, diagnostics, reasoning, etc. Bayes Net is built on probability distribution and relies on the laws of probability for prediction and anomaly detection. Bayes Net supports both discrete and continuous attributes, and it can be defined as follows:P(A,B) = P(A|B)P(B) = P(B|A)P(A) => P(A|B) = P(B|A)P(A)/P(B)(5)

## 4. Experimental Analysis

Stratified 10-fold cross-validation was used during the training phase of the algorithms. It is an evaluation technique that runs repeated percentage splits of data on the model being tested [43]. The considered classification algorithms were used to derive models for the prediction of ASD, and the models were evaluated in this research using accuracy, specificity, and sensitivity rates. For ASD feature–feature analysis, the RStudio heat map was used. The WEKA [44] machine-learning tool was used for running the classification model. None of the hyperparameters for the feature selection and classification methods were changed to specific values; hence, we used the default values.

The results from applying the considered feature-selection techniques (Relief-f, GR) are shown in Table 3. It is worth mentioning that most of the features belonged to Category A of DSM-5 (focus on social communication and interaction across different context), and two features (Q8 and Q10) belonged to Category B of DSM-5 (focus on restricted, repetitive patterns of behavior, interests, or activities) albeit partly for Q8. In addition, it was evident that the attributes that could help detect ASD in toddlers were Q1–Q9, as they appeared before other attributes, such as demographics, achieving the highest-ranking score in each feature-selection method used. However, the top-ranked questions that were reported as per the three feature selection methods were Q3, Q4, Q5, Q6, Q8, and Q9 (see Table 3). For instance, according to the GR and the Pearson correlation, the result showed that the three influential questions were Q6 (0.28/0.59), Q9 (0.26/0.56), and Q5 (0.24/0.54). These questions confirmed deficits in joint attention, social communication, and interaction, and could detect a non-verbal communication problem and the ability of the toddler to pretend to play. However, the Relief-f method reported different results for the second and third rank. The top-three ranked questions were Q6 (0.3), Q5 (0.3) and Q8 (0.28). Question 5 detected a social communication problem that manifested as the toddler’s inability to pretend play. 

All feature-selection methods showed that Q5 and Q6 were strong predictors of autism in toddlers. These questions are related to social communication and interaction, as per the DSM-5, and belong in the same DSM-5 domain. However, the results here were consistent with previous studies conducted in ASD research. Social communication and interaction and delay in language ability are early symptoms that trigger the need for more autism screening [45].

According to the DSM-5, autism diagnostic criteria can be split into two groups of features. One is “persistent deficits in social communication and social interaction” and the other is “restricted, repetitive patterns of behavior, interests, or activities” (Autism Speaks, 2013). The first set of features can be further divided into two distinct features: communication and social interaction. Table 1 earlier shows the breakdown of Q1–Q10 according to DSM-5. The purpose of this research was to identify the features that can help detect ASD in toddlers, so we can break this down further to find the most influential social behaviors according to DSM-5 guidelines. Q1–Q9 were the top features from the selection methods, covering social interaction and communication cognitive behaviors, while Q10 relates to repetitive behaviors and is ranked as the lowest of all QCHAT-10 responses. Table 4 also shows Set 3, which is made up the top five features: Q4, Q5, Q6, Q9, and Q8. All of these features are in the social interaction category, except Q8, which concerns mainly communication and, in some cases, repetitive behaviour (speech). The result for the feature-selection methods enabled us to create four different sets of features for the classification, as shown in Table 4. Social interaction plays the biggest role in assessing toddlers and determining whether they are on the spectrum. 

Classification methods were used on four sets of features to compare the performance of the models and to help us evaluate certain behavioral features. The first set contained a full set of attributes; the second set included Q1–Q9, as they were the common highest-ranked features in all three feature selection results; the third set had the top five attributes; and the last set contained the lowest-scoring attributes. 

We used the LR algorithm for the classification. In addition, the Q-CHAT-10 features were added incrementally to the LR classifier. We started with Q1 against the class, then added Q2. As mentioned, the questions were added incrementally and the changes in the sensitivity were reported after each feature addition. Figure 3 shows the changes in the sensitivity of the LR classifier as the features are added. The reported sensitivity of the model when only Q1 was used was 85.5%. When Q2–Q4 were added to the data with Q1, decreases in sensitivity were recorded (79.70%), indicating the negative effect on the model’s sensitivity. However, this decrease in the sensitivity changed when Q5 was added to the data—an increase was noted and the sensitivity reached 90.9%. It is worth emphasizing that Q5 was one of the three top-ranked features selected by the feature selection methods. 

Figure 4 shows the correlation among the QCHAT-10 attributes only. As shown in the color legend, blue represents negative, while red represents a positive correlation (ranging from −1 to 1). As depicted in Figure 4, Q3 was highly correlated with four questions (Q4, Q9, Q1, and Q5). In addition, it was evident that Q3 and Q4 had the largest correlation of 0.48, followed by Q6 and Q4, which had a correlation of 0.42, and then by Q7 and Q5, which had a correlation of 0.41. Referring to Table 1, both questions measured whether a child was able to express a certain interest by pointing. This could explain the high correlation between Q3 and Q4. In addition, Q6 and Q4 were related—they measured whether a child was looking or pointing to where an adult was looking. Furthermore, Q7 and Q5 measured whether a child showed comfort in playing roles—both questions measured a child’s ability to interact in a given social context. It is clear that only Q7 and Q10 had no correlation (close to 0) with each other. Q10 was also slightly correlated with other features, achieving only 0.13 as the highest correlation with Q4.

The results (sensitivity, specificity, accuracy) of the classification algorithms on the entire set of features can be taken as a point of reference when comparing results of these algorithms on the other subsets of features as shown in Figure 5. The accuracies of the LR and Bayes Net models seem to be consistent when processing datasets 2, 3, and 4, when compared with models derived from dataset 1. In classifying toddlers in terms of ASD, the Bayes Net algorithm derived better accuracy rates against dataset 2 than the models derived from dataset 1. For instance, the Bayes Net algorithm reported 23 cases as false positives and 14 cases as false negatives from Set 1, while the other models reported 15 cases as false positives and 10 cases for false negatives. The Bayes Net and LR algorithms derived models from Set 2 and Set 3 with consistent accuracy, specificity, and sensitivity rates. The two algorithms reported the worst performance result against Set 4. For example, LR reported 133 cases as false negatives and 61 as false positives. Bayes Net reported 59 instances as false positives and 120 instances as false negatives.

## 5. Conclusions, Limitations, and Ethical Implications

This research assessed behavioral features related to the QCHAT-10 ASD medical screening method for toddlers by applying machine-learning techniques. The data were first filtered to obtain toddler data observations, and descriptive analysis was conducted to obtain an understanding of the data before any further testing. GR, Pearson correlation, and Relief-f feature-selection techniques were used to evaluate social, behavioral, communication, and repeated-learning features that could help with the early detection of ASD traits in toddlers. The preliminary feature-selection assessment reported consistent results in terms of the influential features for ASD. It was evident that behavioral traits, such as the social-interaction cognitive behaviors from QCHAT-10 responses, were significant features that could help predict ASD in toddlers. The results showed that Set 3, which is made up of four social features (Q4, Q5, Q6, and Q9) and one communication feature (Q8), was significant in determining ASD during clinical screening. The result for the feature-selection methods allowed us to create three different sets of features for the classification phase. These could be embedded within a digital system for clinicians to use during the clinical assessment of autism. Social interaction played the biggest part in assessing toddlers and determining if they could be classified as having ASD traits, at least based on the machine-learning techniques and the dataset we considered. However, the results showed some overlapping among the features, in which some measured the same domain within the DSM-5, such as Q3 and 4 or Q5 and 7. 

The predictive models derived by the machine-learning algorithms (LR and Bayes Net) from the data showed competitive performance in classifying toddlers. Particularly, when processing Set 1, which contained all features, the derived models provided high accuracy, specificity, and sensitivity rates via the machine-learning algorithms. In particular, the accuracy rates of the LR and Bayes Net algorithms, derived from the behavioral data subsets (2,3), were within acceptable medical standard rates, reaching up to 95% accuracy by LR algorithm. In addition, the models derived by the machine-learning algorithm from Subset 3 (the minimal data subset) showed competitive performance for sensitivity rates, reaching up to 89% by the Bayes Net algorithm. This demonstrated that processing only five behavioral features related to toddlers by data-driven algorithms produced competitive performance in terms of accuracy, specificity, and sensitivity measurements.

A limitation of this is study was the low number of data observations that were used for toddlers in generating the models. In addition, the study was restricted to the machine-learning and feature-selection techniques that were used, so including other techniques, especially advanced AI methods such as deep learning that may consider other types of features related to eye-tracking, may be a way forward in the future. In the near future, we plan to include more data observations and expand on the techniques used. This will allow a more comprehensive conclusion about how feature selection can impact the detection of ASD, especially when including complex features related to play or social interaction from videos, as well as the movements of toddlers. 

The application of AI and machine learning for ASD classification presents a variety of ethical implications that require careful consideration. While these technologies hold promise in enhancing early detection and improving personalized interventions for individuals on the spectrum, they also raise concerns that revolve around privacy, bias, and informed consent. One of the ethical concerns for individuals with ASD pertains to their privacy. AI systems often require vast amounts of data for training the algorithms, including personal information about patients. Ensuring the security and proper handling of these data becomes crucial to prevent unauthorized access. 

Moreover, bias in classification models is an ethical challenge. If the training data is not diverse and sufficiently representative, the resulting models might reflect bias and could lead to inaccuracies in diagnosis. Efforts must be made to mitigate bias and ensure fairness in the development and deployment of these systems. Finally, informed consent becomes a critical issue when using AI and machine-learning techniques for ASD classification. Individuals and their families should have a clear understanding of how their data will be used, the potential benefits, and the limitations of automated-based assessments.

## Figures and Tables

**Figure 1 bioengineering-10-01131-f001:**
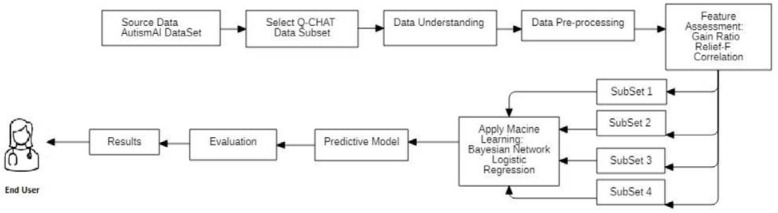
Methodology Followed.

**Figure 2 bioengineering-10-01131-f002:**
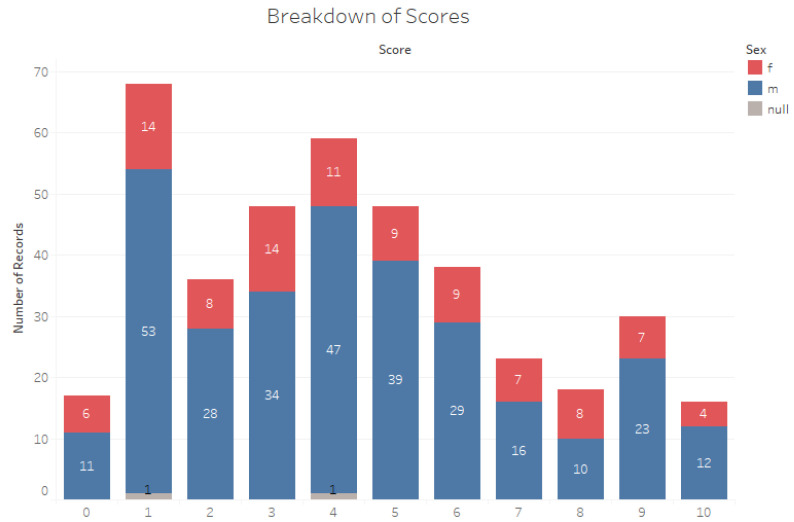
Breakdown of scores by males and females.

**Figure 3 bioengineering-10-01131-f003:**
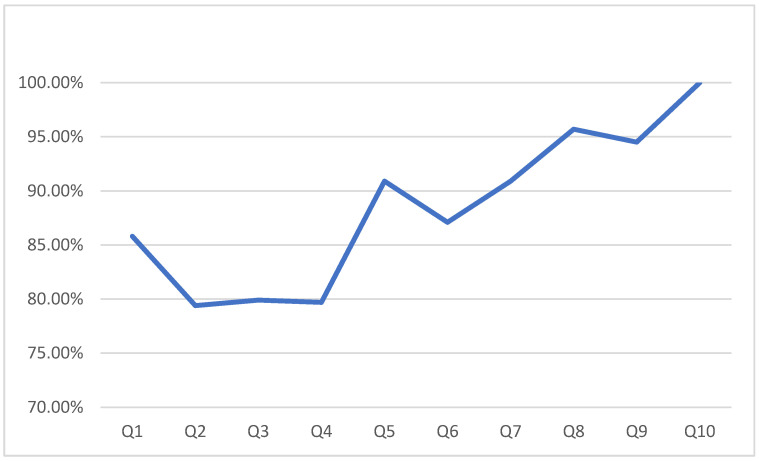
Sensitivity changes as attributes are added.

**Figure 4 bioengineering-10-01131-f004:**
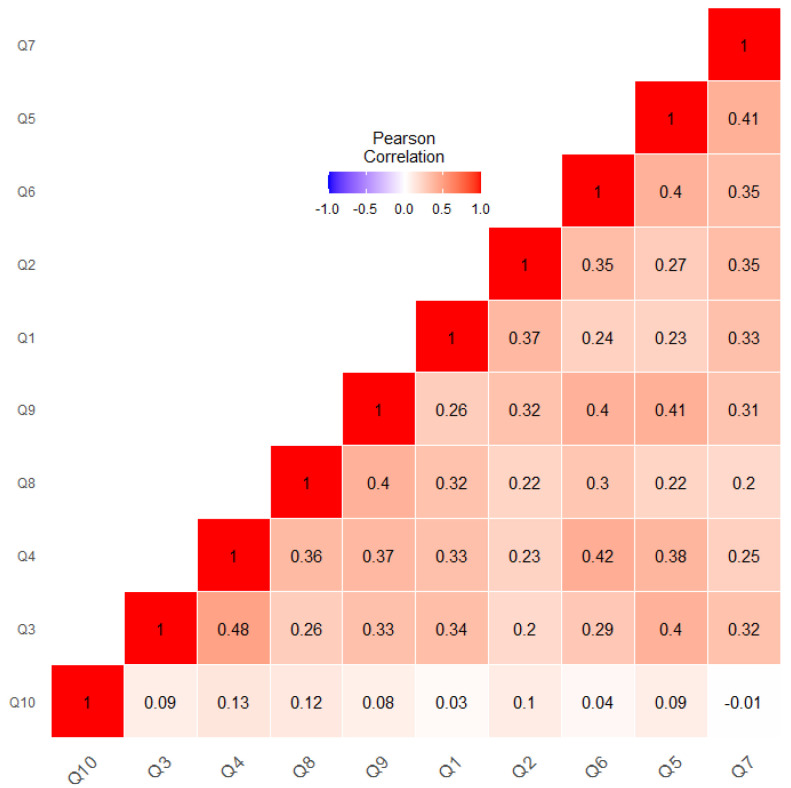
Feature-to-feature correlation for QCHAT-10 attributes.

**Figure 5 bioengineering-10-01131-f005:**
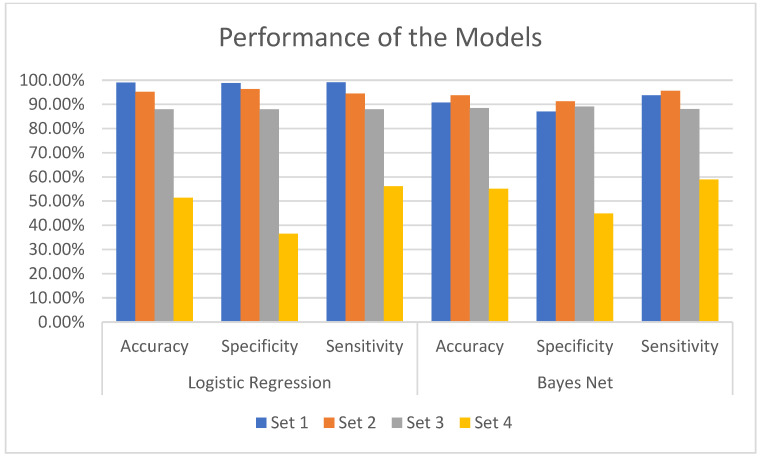
Accuracy, specificity, and sensitivity rates derived from different data subsets.

**Table 1 bioengineering-10-01131-t001:** Literature review summary.

Methods Used	Data Used	Performance	Reference
DLiB library Deep learning Kaldi toolkit	125 toddlers	Sensitivity: 80.00% Specificity: 69.80% Accuracy: 74.80%.	[10]
LR RF	IBM MarketScan Health Claims database; 38,576 observations	RF: - AUCROC: 78.00% - Specificity: 93.00% LR: - AUCROC: 76.00% - Specificity: 90.00%	[11]
CNN	Dataset: collected by ASDTests 6075 observations	- Accuracy: 95.53%, - Sensitivity: 97.63% - Specificity: 98.63%	[12]
SOM RF NB	2000 observations	NB: - Accuracy: 93.00%, - Precision: 93.00% - Recall: 94.00% RF: - Accuracy: 96.00%, - Precision: 96.00%, - Recall: 97.00%	[13]
C4.5 RIPPER RF NB	ASDTest dataset; 1054 toddler data observations	No data sampling: NB: Sensitivity: 96.20% C4.5: Sensitivity: 92.30% RIPPER: Sensitivity: 92.40% RF: Sensitivity: 95.30%	[15]
Symmetrical uncertainty (SU), IG, fast-correlated-based filter (FCBF), leave one out cross-validation (LOOCV), gini index, and chi-square ID3 ADABoost Knn	ASDTest dataset; 1054 toddler data observations	No data sampling. Sensitivity rates between 93% and 98%, depending on the feature sets used by the classification algorithm. The highest sensitivity rate was achieved by ADABoost.	[18]
NB with data sampling: SMOTE RUS	ASDTest dataset; 1118 adult data observations	SMOTE+ NB: - Sensitivity: 96.00% RUS + NB: - Sensitivity: 94.00% No sampling + NB: - Sensitivity: 93.00%	[22]
mRMR and chi-square testing feature selection C4.5 RIPPER RF NB SVM	ASDTest dataset; 1054 toddler data observations	No data sampling. Sensitivity rates between 93% and 97.50%, depending on the feature sets used by the classification algorithm. The highest Sensitivity rate was achieved by the SVM.	[17]
NB with data sampling: SMOTE ROS RUS	ASDTest dataset; over 1000 observations	SMOTE + NB: - Sensitivity: 95.00% - Specificity: 94.00% - Precision: 95.90% ROS + NB: - Sensitivity: 94.20% - Specificity: 96.45% - Precision: 94.30% RUS + NB: - Sensitivity: 94.30% - Specificity: 96.09% - Precision: 94.3%0	[14]
kNN, SVM RF	ASD dataset of UCI machine-learning data repository	kNN: Accuracy: 95.70% Sensitivity: 95.15% F-measure: 94.64% AUC: 96.00% SVM: Accuracy: 99.90% Sensitivity: 99.90% F-measure: 99.90% AUC: 100% RF: Accuracy: 99.90% Sensitivity: 99.90% F-measure: 99.90% AUC: 99.90%	[16]
Multilayer perceptron (MLP) classifier	Social responsiveness scale (SRS) [28] - child/adolescent questionnaire; 16,527 children/adolescents	AUC: 92.00%	[27]
SVM CNN ANN	An integrated data from the UCI machine-learning data repository, consisting of three datasets with 20 common attributes	CNN algorithm. - Accuracy: 98.30% SVM: - Accuracy: 97.95% ANN: - Accuracy: 97.60%	[29]
Multivariate LR MLP RF	EMR data from a single Israeli health maintenance organization; 96,138 EMR children information	Multivariate LR: - Accuracy: 94.90% - AUC: 72.70% MLP: - Accuracy: 95.50% - AUC: 70.00% RF - Accuracy: 96.50% - AUC: 69.30%	[30]

**Table 2 bioengineering-10-01131-t002:** Mapping Q-CHAT-10 items to the corresponding DSM-5 domains.

Question Number	Question Details	Corresponding DSM-5
Q1	Does your child look at you when you call his/her name?	Deficits in social communication and interaction (problem with social initiation and response)
Q2	How easy is it for you to have eye contact with your child?	Deficits in social communication and interaction (non-verbal communication problem)
Q3	Does your child point to indicate that s/he wants something (e.g., a toy that is out of reach)?	Deficits in joint attention and social communication and interaction (non-verbal communication problem)
Q4	Does your child point to share interest with you (e.g., pointing at an interesting sight)?	Deficits in joint attention and social communication and interaction (non-verbal-communication problems)
Q5	Does your child pretend (e.g., care for dolls, talk on a toy phone)?	Deficits in social communication and interaction related to pretend play
Q6	Does your child follow where you are looking?	Deficits in joint attention and social communication and interaction (non-verbal communication problems)
Q7	If you or someone else in the family is visibly upset, does your child show signs of wanting to comfort them (e.g., stroking their hair, hugging them)?	Deficits in social communication and interaction (problems with social initiation and response)
Q8	Would you describe your child’s first words as (Very typical, Quite typical, Slightly unusual, Very unusual, My child doesn’t speak)	Deficits in social communication and interaction related to language development. Stereotyped/repetitive speech
Q9	Does your child use simple gestures (e.g., wave goodbye)?	Deficits in social communication and interaction (non-verbal communication problem)
Q10	Does your child stare at nothing with no apparent purpose?	Shows restricted and repetitive patterns of behavior, interests, or activities (stereotyped behaviors)

**Table 3 bioengineering-10-01131-t003:** Feature selection results.

Attribute Rank	Gain Ratio Score	Attribute Rank	Pearson Correlation Score	Attribute Rank	Relief Score
Q6	0.281297	Q6	0.5978	Q6	0.30551
Q9	0.263325	Q9	0.5653	Q5	0.30501
Q5	0.240884	Q5	0.5492	Q8	0.28446
Q4	0.222894	Q4	0.5229	Q9	0.28521
Q3	0.222216	Q8	0.5163	Q4	0.24612
Q8	0.20949	Q7	0.4805	Q3	0.20902
Q7	0.182863	Q3	0.4783	Q2	0.19799
Q2	0.163217	Q2	0.461	Q7	0.19098
Q1	0.146544	Q1	0.4181	Q1	0.17945
FamilyASDHistory	0.013727	FamilyASDHistory	0.1316	Q10	0.05514
Ethnicity	0.006148	Age	0.1201	Ethnicity	0.02957
Jaundice	0.006557	Jaundice	0.0903	FamilyASDHistory	0.02331
User	0.005728	User	0.0802	User	0.01805
Q10	0.000547	Q10	0.0272	Age	0.00827
Sex	0.000169	Ethnicity	0.0252	Jaundice	0.001
Age	0	Sex	0.0137	Sex	-0.00802

**Table 4 bioengineering-10-01131-t004:** Feature sets to be tested by classification.

Set 1: No Feature Selection	Set 2: Q1 to Q9	Set 3: Highest Scoring attributes	Set 4: Secondary/Lowest Scoring Attributes
Q1 (Communication)	Q1	Q6	FamilyASDHistory
Q2 (Social interaction)	Q2	Q9	Age
Q3 (Communication)	Q3	Q5	Q10
Q4 (Social interaction)	Q4	Q8	User
Q5 (Social interaction)	Q5	Q4	Jaundice
Q6 (Social interaction)	Q6		Ethnicity
Q7 (Social interaction)	Q7		Sex
Q8 (Communication)	Q8		
Q9 (Social interaction)	Q9		
Q10(Repetitive patterns)			
Age			
Sex			
Ethnicity			
Jaundice			
Family ASD history			
User			

## Data Availability

Data are available on Kaggle data repository & originally generated by AutismAI’s authors.

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
