# Peer review of "Assessing Autistic Traits in Toddlers Using a Data-Driven Approach with DSM-5 Mapping"

_bioengineering, 2023, doi:10.3390/bioengineering10101131_

Round 1

Reviewer 1 Report

Interesting work, I would like to thank the authors for this contribution. The study presents an interesting ML application of in the autism context. Additionally, I commend the authors for acknowledging potential limitations. However, I would like to offer some suggestions for improvement to the next version, please.

(1)

The literature review demonstrates a commendable effort. However, it would be valuable to expand the discussion on eye-tracking methods coupled with ML for autism diagnosis. Eye-tracking is particularly one of the widely used approaches in the context of autism diagnosis. One possible example to mention:

https://doi.org/10.2196/27706

(2)

It was not mentioned whether the study obtained some Institutional Review Board (IRB). Please mention that clearly.

(3)

It’s worth discussing further why the study considered the LR and Bayes approach for developing the ML models. Also, did the authors explore other models?

(4)

Please discuss possible ethical considerations related to using data from toddlers or the potential implications of using AI techniques for diagnosing ASD. Addressing ethical concerns and potential biases in the models is important for responsible research.

(5)

For the future work, it's important to consider assessing the model's performance on external datasets to validate its effectiveness across different sources of data.

(6)

Please mention all the libraries used to develop the ML models, and cite their references.

Overall, I appreciate the authors' efforts and look forward to seeing an improved version of this study.

Reviewer 2 Report

The work is good. However, I have some major concerns to resolve before acceptance:

1-Abstract would be enhanced by adding performance measure scores. Also, please highlight your findings.

2- Please check the citations and references style; it looks like they are incorrect throughout the manuscript.

3- In line 43, typo error: such as attention deficit hyperactivity disorder [ADHD]

4- The introduction section is too large and needs to be concise.

5- Please highlight your contributions in the introduction section. 

8- In related work, please mention the gap you identified from previous studies.

7-The related work section would be enhanced by citing advanced deep learning-based classification studies such as:

Rehman, Amjad, et al. "Transfer Learning-Based Smart Features Engineering for Osteoarthritis Diagnosis from Knee X-ray Images." IEEE Access (2023).

Raza, Ali, Kashif Munir, and Mubarak Almutairi. "A novel deep learning approach for deepfake image detection." Applied Sciences 12.19 (2022): 9820.

8-The textual content in figure 1 is challenging to read, and the components within the figure lack proper alignment. Consequently, it is advisable to consider the utilization of professional diagramming tools, such as app.diagrams.net, for the creation of figures, which would likely enhance both readability and visual coherence.

9-How can you say your newly created dataset is authentic? Also, I think there is a human error in your data. How you validate the authenticity of data. Provide a justification.

10-The methodology section missing alot of parts. Like what steps are performed during data preprocessing and how feature selection is performed. Provide a step-by-step workflow diagram in the Methodology section that would be better to understand the proposed methodology.

11-The equation numbers are not aligned. See line number 314....

12-In Figure 2, what is meant by the null label?

13-In Figure 5, you achieved 99 accuracy. It looks overfit. Why have not results of k-fold cross validation are evaluated?

14- There is no comparison with state-of-the-art approaches is performed. Please add state-of-the-art techniques and comparisons

15-The manuscript is missing future work. Please create a subsection for future work.

16-Please proofread the manuscript. I found many typos and grammatical errors. There is a need for extensive English revision.

Extensive editing of English language required

Round 2

Reviewer 1 Report

Thanks, I have no further comments.

Reviewer 2 Report

Authors have solved all issues raised by me so accepted. Thanks

Accepted